# Impact of Extended Use of Ablation Techniques in Cirrhotic Patients with Hepatocellular Carcinoma: A Cost-Effectiveness Analysis

**DOI:** 10.3390/cancers14112634

**Published:** 2022-05-26

**Authors:** Toulsie Ramtohul, Valérie Vilgrain, Olivier Soubrane, Mohamed Bouattour, Alain Luciani, Hicham Kobeiter, Sébastien Mule, Vania Tacher, Alexis Laurent, Giuliana Amaddeo, Hélène Regnault, Julie Bulsei, Jean-Charles Nault, Pierre Nahon, Isabelle Durand-Zaleski, Olivier Seror

**Affiliations:** 1AP-HP, Health Economics Research Unit, 75004 Paris, France; toulsie@me.com (T.R.); julie.bulsei@aphp.fr (J.B.); isabelle.durand-zaleski@aphp.fr (I.D.-Z.); 2AP-HP, Department of Radiology, Jean Verdier Hospital, 93140 Bondy, France; 3AP-HP, Department of Radiology, Beaujon Hospital, 92110 Clichy, France; valerie.vilgrain@aphp.fr; 4INSERM U1149, Centre de Recherche Biomédicale Bichat-Beaujon, CRB3, 75018 Paris, France; 5AP-HP, Department of HPB Surgery and Liver Transplantation, Beaujon Hospital, 92110 Clichy, France; olivier.soubrane@aphp.fr; 6AP-HP, Department of Digestive Oncology, Beaujon Hospital, 92110 Clichy, France; mohamed.bouattour@aphp.fr; 7INSERM IMRB Unit U 955, Equipe 18, 94010 Créteil, France; alain.luciani@aphp.fr (A.L.); sebastien.mule@aphp.fr (S.M.); vania.tacher@aphp.fr (V.T.); alexis.laurent@aphp.fr (A.L.); giuliana.amaddeo@aphp.fr (G.A.); helene.regnault@aphp.fr (H.R.); 8AP-HP, Department of Radiology, Henri Mondor Hospital, 94000 Créteil, France; hicham.kobeiter@aphp.fr; 9AP-HP, Department of Liver Surgery, Henri Mondor Hospital, 94000 Créteil, France; 10AP-HP, Department of Hepatology, Henri Mondor Hospital, 94000 Créteil, France; 11AP-HP, Department of Hepatology, Jean Verdier Hospital, 93140 Bondy, France; jean-charles.nault@aphp.fr (J.-C.N.); pierre.nahon@aphp.fr (P.N.); 12Unité Mixte de Recherche 1162, Génomique Fonctionnelle des Tumeurs Solides, Institut National de la Santé et de la Recherche Médicale, 75010 Paris, France; 13French League Against Cancer, Education and Research in Health Medicine and Human Biology, University Paris 13, Sorbonne Paris Cité, 75005 Paris, France; 14ECEVE, UMRS 1123, French National Institute of Health and Medical Research, 75010 Paris, France; 15AP-HP, Department of Public Health, Henri Mondor Hospital, 94000 Creteil, France

**Keywords:** cost-effectiveness, hepatocellular carcinoma, percutaneous ablation, TACE, treatment management

## Abstract

**Simple Summary:**

The optimal management of non-metastatic hepatocellular carcinoma (HCC) remains debated. The association between HCC and cirrhosis influences prognosis and therapeutic choices between curative and palliative treatments. The goal of our retrospective study was to evaluate the cost-effectiveness of the extended use of ablation for the treatment of HCC with cirrhosis in an expert ablation center when compared to the non-extended use of ablation in equivalent tertiary care centers. In a propensity-score matched cohort of 532 patients with naïve HCC, the extended use of ablation led to better compliance with the Barcelona Clinic Liver Classification (BCLC) guidelines (80% vs. 67%) and was more effective and less expensive than the non-extended use of ablation strategy, particularly at an earlier stage of the disease. The shift from curative to palliative treatments was noted in a considerable percentage of patients; therefore, this needs to be redefined as the wide choice of ablation techniques and technical advances in imaging guidance increase the curative options available to treat a maximum of patients with HCC.

**Abstract:**

Background: To evaluate the cost-effectiveness of the extended use of ablation for the treatment of hepatocellular carcinoma (HCC) with cirrhosis in an expert ablation center when compared to the non-extended use of ablation in equivalent tertiary care centers. Methods: Consecutive cirrhotic patients with non-metastatic HCC, no prior treatment, and referred to three tertiary care centers between 2012 and 2016 were retrospectively identified. The Bondy group, including all of the patients treated at Jean Verdier Hospital, where the extended use of ablation is routinely performed, was compared to the standard of care (SOC) group, including all of the patients treated at the Beaujon and Mondor Hospitals, using propensity score matching. A cost-effectiveness analysis was carried out from the perspective of French health insurance using a Markov model on a lifetime horizon. Results: 532 patients were matched. The Bondy group led to incremental discounted lifetime effects of 0.8 life-years gained (LYG) (95% confidence interval: 0.4, 1.3) and a decrease in lifetime costs of EUR 7288 (USD 8016) (95% confidence interval: EUR 5730 [USD 6303], EUR 10,620 [USD 11,682]) per patient, compared with the SOC group, resulting in a dominant mean incremental cost-effectiveness ratio (ICER). A compliance with the Barcelona Clinic Liver Classification (BCLC) guidelines for earlier stage contributed to the greater part of the ICER. Conclusion: The extended use of ablation in cirrhotic patients with HCC was more effective and less expensive than the non-extended use of the ablation strategy.

## 1. Introduction

The optimal management of non-metastatic hepatocellular carcinoma (HCC) remains debated. The association between HCC and cirrhosis influences prognosis and therapeutic choices between surgical and nonsurgical treatments [1,2]. While liver transplantation offers the best survival outcomes in highly selected patients [3], it is limited by the shortage of organs and tumor progression while the patient is on the waiting list [4]. Hepatic resection has demonstrated comparable results following upfront liver transplantation in small solitary HCC with compensated cirrhosis [5]. Unfortunately, at diagnosis, a majority of patients are not candidates for surgical treatment because of contraindications, mostly represented by (even slightly) impaired liver function, portal hypertension, or the presence of comorbidities [6]. These limitations have been highlighted by the prospective follow-up of European compensated cirrhotic patients included in HCC surveillance programs [7,8]. According to the Barcelona Clinic Liver Classification (BCLC), percutaneous ablation is proposed as the first-line option for unresectable early-stage HCC with up to three nodules smaller than 3 cm not amenable to liver transplantation [9]. However, the BCLC decisional algorithm does not address the numerous technical issues that can limit the feasibility of ablation in patients bearing early and even very early stages of HCC. Thus, in retrospective real-life studies, radiofrequency ablation procedures (RFA) have been denied in up to 30% of cases, with a negative impact on survival rates because of technical issues or a high-risk location of the tumor [10], especially in early-stage HCC patients who are a theoretically good candidate for ablation but receive, according to the migration stage strategy, suboptimal palliative endovascular treatment, mainly trans-arterial chemoembolization (TACE) [11,12]. On the opposite side of the BCLC algorithm, it has been advocated that ablation could improve the outcomes of a subset of patients with intermediate and even advanced stages usually referred for TACE, TARE, or systemic treatments [13,14,15,16]. Whether to reduce resorting to the migration stage strategy for the treatment of early stages or to treat more advanced stages of HCC by ablation, the use of expert techniques and technologies such as imaging fusion, artificial pleural effusion or ascites, microwave, multi-bipolar RFA, or irreversible electroporation (IRE) are mandatory [17,18,19,20].

For several years, all of these means have been routinely used at the Jean Verdier University Hospital (Bondy, France) to extend the use of percutaneous ablation for the treatment of a subset of HCC patients commonly treated with endoarterial treatments either by applying a migration stage strategy or in accordance with the BCLC recommendations for more advanced stages (Figure 1) [19,20]. The main goal of this study was to evaluate the cost-effectiveness of the extended use of percutaneous ablation for the treatment of HCC with cirrhosis in an expert center employing cutting-edge ablation techniques compared to the non-extended use of ablation in equivalent tertiary care centers.

## 2. Results

### 2.1. Study Population

Between 1 January 2012 and 1 February 2016, 832 HCC patients received curative or palliative modalities as first-line treatment in all three centers. Of these, three were excluded because of decompensated cirrhosis and hepatocholangiocarcinoma. Two hundred and seventy-two patients were treated at Jean Verdier University Hospital and 557 patients at Beaujon and Henri Mondor Hospitals (Figure 2). After propensity score matching, the final cohort comprised 532 patients. The two groups had similar baseline characteristics and BCLC stages (Table 1). Cirrhosis was present in all of the patients.

### 2.2. Allocation of First-Line Treatments

The distribution of the first-line treatments between the two groups differed significantly (Figure 3), with more curative treatments being delivered in the Bondy group (71% vs. 37%; *p* < 0.001). In the Bondy group, 188 patients (71%) underwent percutaneous ablation, one patient (<1%) a resection, 47 patients (18%) TACE, and 30 (11%) received Sorafenib as first-line therapy. There were 21 (8%) and 11 (4%) ablated patients classified as being at the intermediate and advanced stages. For the very early/early, intermediate, and advanced stages, there were 156 (90%), 31 (56%), and 25 (66%), respectively, first-line treatments adherent to BCLC guidelines (overall adherence: 80%). In the SOC group, 35 patients (13%) were treated with hepatic resection, 64 patients (24%) by percutaneous ablation, 128 patients (48%) by TACE, and 39 (15%) by Sorafenib. For the very early/early, intermediate, and advanced stages, there were 92 (56%), 50 (86%), and 35 (81%), respectively, first-line treatments adherent to BCLC guidelines (overall adherence: 67% vs. 80%, *p* = 0.004). Overall, the palliative treatment stage migration accounted for 21 patients (8%) vs. 76 patients (29%) in the Bondy and SOC group (*p* < 0.001), respectively.

### 2.3. Therapeutic Trajectories after First-Line Treatments

The number of patients treated with curative modalities after first-line treatments was greater in the Bondy group vs. the SOC group (59% vs. 27%, *p* < 0.001), regardless of the curative or palliative first-line treatment (Figure 4). Among the 188 patients (71%) who were given ablation in the Bondy group, ablation was the most common consecutive strategy, and 133 patients (71%) received one or more consecutive ablation therapies. Whereas among the 99 patients (37%) who were treated by either first-line hepatic resection or ablation in the SOC group, TACE was the most frequent consecutive procedure and 32 patients (32%) received one on more TACE. When TACE was performed as first-line treatment, even for earlier stage HCC, the number of patients receiving consecutive curative treatment was low in both groups. Among the 49 patients treated with first-line TACE in the Bondy group, 15 patients were finally treated with ablation (31%). Among the 128 patients treated with first-line TACE in the SOC group, nine patients were finally treated by hepatic resection (7%), 18 by ablation (14%), and 19 by liver transplantation (15%). Overall, 42 patients (8%) underwent liver transplantation, including 12 (5%) in the Bondy group (12 after first-line ablation) and 30 (11%) in the SOC group (19 after first-line TACE, four after first-line ablation, and seven after first-line hepatic resection). Eighty-one percent of liver transplants were for earlier stage HCC in both groups.

### 2.4. Overall Survival Analysis 

The median follow-up was 47 months (IQR 34-62). In the Bondy group, the median overall survival was 35 months (IQR 14-58), 12 months (IQR 6-20), and 4 months (IQR 1-6) with ablation, TACE, and Sorafenib, respectively. In the SOC group, the median overall survival was 55 months (IQR 8-not estimable), 37 months (IQR 13-not estimable), 18 months (IQR 10-35), and 9 months (IQR 4-22) with hepatic resection, ablation, TACE, and Sorafenib, respectively (Appendix A). Among the very early and early HCC patients, first-line curative treatments (hepatic resection and ablation) did not display any statistically significant difference in terms of OS between the two groups (median OS 38 vs. 37 months in the Bondy and SOC groups, respectively; *p* = 0.96). However, in very early and early HCC patients, the overall survival was significantly shorter among those receiving first-line TACE in the SOC group than in patients receiving ablation in the Bondy group (median OS 18 vs. 38 months; *p* = 0.001). In the patients with intermediate and advanced HCC, there was a trend towards better overall survival among those undergoing ablation techniques than receiving TACE (median OS 26 vs. 19 months; *p* = 0.45) or Sorafenib (median OS 13 vs. 9 months; *p* = 0.35) (Figure 5).

### 2.5. Cost Effectiveness Analysis 

The model estimated the average lifetime costs per patient as being EUR 18,205 (USD 20,026) and EUR 25,493 (USD 28,043) in the Bondy and SOC groups, respectively. Extrapolated survival reached 12.0 years in the Bondy group and 11.2 years in the SOC group (Table 2). The Bondy group led to incremental discounted lifetime effects of 0.8 LYG (95% confidence interval: 0.4, 1.3) and a decrease in lifetime cost by EUR 7288 (USD 8017) (95% confidence interval: EUR 5730 [USD 6303], EUR 10,620 [USD 11,682]) per patient, compared with the SOC group, resulting in a dominant mean incremental cost-effectiveness ratio. Compliance with the BCLC guidelines for very early and early-stage disease contributed to the greater part of the ICER (Figure 6). Among the patients with early-stage HCC treated by ablation (Bondy group) vs. TACE (SOC group), the Bondy group resulted in a lifetime gain of 5.8 LYGs per patient (95% confidence interval: 5.2, 6.5) and a decrease in cost by EUR 8778 (USD 9656) (95% confidence interval: EUR 5422 [USD 5964], EUR 12,804 [USD 14,084]). The deterministic sensitivity analysis found that the overall results were most sensitive to the probability of success of ablation in the Bondy group, i.e., the ability of the technique to provide actual curative treatment. Monte Carlo simulations found that the Bondy group strategy was dominant in 96.2% of the simulations. With a restricted 5-year horizon, the Bondy group strategy also dominated the SOC group (The results for all sensitivity analyses are presented in the Appendix A).

## 3. Discussion

The extended use of percutaneous ablation beyond common feasibility issues (ultrasonography tumor invisibility, at-risk location, large ablation, thrombocytopenia, etc.) within the very early and early stages commonly leads to the application of migration-stage strategies and selected intermediate and advanced stages for the treatment of HCC with cirrhosis led to lower healthcare costs and more life-years gained on a lifetime horizon, for a high proportion of the iterations (96.2%), compared to the non-extended use of ablation strategy. Patients with early-stage disease accounted for the majority of patients in the cohort. The larger proportion of earlier-stage HCC in this cohort may reflect the cost-effectiveness of the screening of the at-risk cirrhotic population, allowing for the application of curative treatments [21]. As a result of the extended use of ablation, the proportion of patients triaged to first-line curative therapies was high in the Bondy group (71%), whereas the treatment allocation in the highly specialized centers of the SOC group revealed an over-use of palliative approaches mainly with TACE (37%). For earlier stages, palliative treatment stage migration accounted for 44% of the patients in the SOC group, representing the most common procedure delivered in the SOC group. These figures seem representative of the patterns of HCC management worldwide since the global HCC BRIDGE study on 18,000 patients found that, across all BCLC stages, TACE was the most frequent first-line HCC treatment in North America, Europe, South Korea, and China [22]. As with the findings of other studies [23,24,25], the overall survival was shorter in early-stage patients treated with suboptimal palliative TACE rather than curative modalities in accordance with the migration stage strategy principle. For cirrhotic patients with frequent surgical contraindications, the ablation strategy appeared to be a credible approach to keep earlier stage HCC in a curative intent with curative techniques. The extended use of ablation, therefore, led to an 80% adherence to BCLC therapeutic management guidelines across all stages and even reached 90% for earlier-stage HCC. Compliance with BCLC guidelines for very early and early-stage disease contributed to the greater part of the ICER with the highest decrease in costs (EUR 8778 [USD 9656]) and the highest increase in lifetime gain per patient (5.8 LYG) and was shown to be robust in the sensitivity analysis (Figure 6). Broadened access to ablation for earlier stage HCC, which appeared to be the cheapest treatment, reduced not only the number of palliative first-line procedures but also enabled better overall survival with fewer expenditures for consecutive treatments with palliative modalities.

The first-line treatment appeared crucial in the management’s trajectory of the patient. For earlier stage HCC treated with first-line TACE, the number of consecutive treatments with ablation, hepatic resection, or liver transplantation only occurred in a very limited number of patients in the SOC group. This finding suggests that such a “wait and see” strategy confers a high risk of therapeutic limitation as most patients will not successfully access liver transplantation. Indeed, these patients are considered at baseline in an “intention-to-transplant” context, which usually recommends TACE as first-line therapy. Given the high proportion of dropouts, such patients should instead be treated by ablation using advanced techniques if necessary to ensure first-line curative management, while liver transplantation could be considered as a salvage option in the event of recurrence [26].

Promoting access to expert skills or centers specialized in ablation techniques might improve prognosis in a large subset of HCC patients, who might otherwise suffer from insufficient application of curative procedures, and reduce the treatment and follow-up costs for the Health Insurance. The Bondy strategy was made possible by technical advances in equipment and the organization of referral networks designed to advise physicians on the feasibility of ablation in a timely manner, thus enabling the optimal management for patients from all parts of the region and reducing inequality regarding their access to curative options [27].

This study had some weaknesses. One limitation concerns its retrospective design, as this could lead to selection bias without randomization. However, we simulated randomization by performing propensity score matching on a large range of separate confounding factors. Furthermore, the cutting-edge technologies used by these tertiary centers might not be available worldwide (e.g., CBCT virtual target display or IRE), which could hinder the extrapolation of the findings [28]. The role of TARE is still under investigation and does not yet form part of the armamentarium for HCC management with PVTT according to the BCLC classification, although it has proved to produce clinically meaningful response rates and prolonged duration of response in patients with solitary unresectable HCC up to 8 cm [29]. The official social health insurance tariffs applied for ablation procedures in France may underestimate the cost of expensive procedures such as complex multi-bipolar radiofrequency or irreversible electroporation. Although the costs of treatments for liver diseases in France are lower than those in the US [30], the respective proportions between treatment options are generally preserved, which suggests that the monetary benefit might be higher for US payers. The net health benefit would be proportional to the current uptake of percutaneous ablation in the country.

## 4. Materials and Methods

### 4.1. Patient Selection

Consecutive cases of HCC, with no prior HCC treatment, diagnosed between 1st January 2012 and 1st February 2016 at the three centers were included. These expert centers treat one-sixth of all in-patients for HCC in France (13,701 incidence of HCC which received a curative or palliative treatment during the 2009–2012 period in mainland France) [30]. They maintain a prospective database of HCC patients (Ark-Dos) and hold weekly HCC multidisciplinary boards attended by hepatologists, oncologists, interventional radiologists, hepatobiliary specialists, and liver transplant surgeons. Only patients with cirrhosis (defined by typical clinical and radiological criteria with or without histological criteria) at the time of the HCC diagnosis were included in the present analyses. The diagnosis of HCC was based on either histological findings or AASLD imaging criteria [31,32]. Portal hypertension was based on gastroesophageal varices, splenomegaly with a platelet count lower than 100,000/mL, ascites, or hepatic venous pressure gradients above 10 mmHg when available. Patient follow-up was recorded until death or the last available clinical information. Patients with extra-hepatic metastases, histological evidence of cholangiocarcinoma, and decompensated cirrhosis classified as Child C were excluded. All of the patients treated at Jean Verdier Hospital, where the extended use of percutaneous ablation is routinely performed, were included in the Bondy group. The so-called “extended use of percutaneous ablation” for the Bondy groups refers to its leading strategy consisting of maximizing the usage ablation in HCC patients within well-admitted indications of the very early and early stages of BCLC by reducing as much as possible the technical contraindications which commonly lead to the application of migration stage principles (i.e., small tumor in dangerous locations or poorly visible using classic guidance imaging means); and in a subset of more advanced stages commonly referred to TACE or systemic treatment (Figure 1). This strategy has been enabled by the continuous efforts of the group for over more than twenty years of involvement in the management of HCC in order to assimilate several advanced techniques and technologies for ablations and imaging guidance into routine practice [13,14,15,16,27,33,34,35,36,37,38,39]. All of the patients treated at Beaujon and Henri Mondor Hospitals were included in the Standard of Care (SOC) group. Because this study was based on an analysis of existing administrative and clinical data, the requirement to obtain informed patient consent was waived by the Institutional Review Board.

### 4.2. Outcome

Overall survival was defined as the interval between initiation of the first treatment and the patient’s death or most recent follow-up visit.

### 4.3. Model Structure

A Markov-based model was developed to extrapolate the short and long-term results. The model described the management of cirrhotic patients with non-metastatic HCC (the Markov structure, underlying assumptions, and calibration are presented in the Appendix A). Four Markov models were therefore built to describe care pathways and outcomes of resection, ablation, TACE, and Sorafenib. After one of the four initial treatments, consecutive therapies could be managed by curative options (salvage transplantation, resection, or ablation) or palliative options (TACE, TARE, and Sorafenib). Multimodal therapies were also considered in the model structure (i.e., ablation with TACE). HCC patients awaiting liver transplantation and treated with a bridge or down-staging therapies prior to liver transplantation were also included. TARE and Sorafenib were grouped together. Deaths from both hepatic and non-hepatic causes were the absorbing state. Health states were common to both groups, and the probabilities of a transition between health states were derived from the Bondy and SOC groups [40]. Patient-level data were used to fit overall survival curves [41]. Survival functions (i.e., Weibull, exponential, and Gompertz) were tested, and the best fit was determined using the lower Bayesian Information Criterion. An exponential distribution was therefore employed. The cycle duration was three months to capture sufficient clinical events. The model was run on a life-time horizon (20 years).

### 4.4. Health Costs

All of the nationally-based direct medical costs of liver resection, percutaneous ablation, liver transplantation, intra-arterial embolization, and Sorafenib treatments were estimated from the perspective of French Health Insurance and were similar in both groups [42]. The data associated with inpatient care were extracted from the discharge summaries using diagnosis-related groups (DRGs) and valued from the national DRG cost study. Pre-operative and follow-up examinations were obtained from the claims databases (Appendix A). The costs of Sorafenib and other medications were obtained from the list prices (red book). The costs were valued in euros, with U.S. dollars (USD) in parentheses (EUR 1 = USD 1.10; 27 May 2020). Outcomes and costs were both discounted at 3% annually to reflect time preference (Table 3).

### 4.5. Cost-Effectiveness Analysis

The economic evaluation estimated costs and life-years gained in the Bondy and SOC groups using the Markov model and calculated an Incremental Cost-Effectiveness Ratio (ICER) in Euros per year of life gained. A one-way deterministic analysis was performed to assess the impact of the following variables: transition probabilities, start age, discount rate, and costs. Probabilistic sensitivity analyses using 1000 Monte Carlo simulations were performed to obtain mean incremental costs and effects with confidence intervals and to evaluate the uncertainty in building cost-effectiveness planes. Gamma distribution was used for cost variables and beta and lognormal distributions for efficacy variables (Appendix A). This report was written in accordance with CHEERS guidelines [43].

### 4.6. Statistical Analysis

The continuous variables were analyzed using the Mann–Whitney and Student tests according to the normality of distributions. Categorical variables were subject to the χ2 test or Fisher’s exact test. In order to compare patients with similar clinical characteristics and prognostic factors in both groups, a one-to-one nearest neighbor propensity score matching procedure with no replacement using an optimal algorithm [44] was performed on the following parameters: age, gender, co-morbidity index, cause of cirrhosis, Child–Pugh score, portal hypertension, alpha-foetoprotein (AFP) level, vascular invasion, number of nodules, size of the largest nodule, and bilobar involvement (Appendix A). A multiple imputation inference method was developed to deal with the missing values [45]. Deaths from any causes were used to calculate the overall survival (OS). The Kaplan–Meier product-limit method was used to plot time to overall survival outcomes, and the comparisons between the curves were performed with the log-rank test. All of the analyzes were conducted using the SAS software application (version 9.4: SAS Institute, Cary, NC, USA). A *p*-value of less than 0.05 was considered to be statistically significant. One thousand Monte Carlo simulations were performed using @Risk Copyright © 2020 Palisade Corporation.

Nationally-based direct medical costs of HCC treatment from French Insurance and for a 3-month cycle were used to define the health state cost mode (fixed value). TACE, trans-arterial chemoembolization.

## 5. Conclusions

This large multicenter study indicates that it is more effective and less expensive to offer cirrhotic patients with HCC an extended use of ablation treatments than a non-extended use of ablation strategy, particularly at the earlier stages of the disease. The shift from curative to palliative treatments was noted in a considerable percentage of patients mainly according to the migration stage principle, which, therefore, needs to be redefined as the wide choice of ablation techniques and technical advances in imaging guidance increase the curative options available to treat a maximum of patients with HCC.

## Figures and Tables

**Figure 1 cancers-14-02634-f001:**
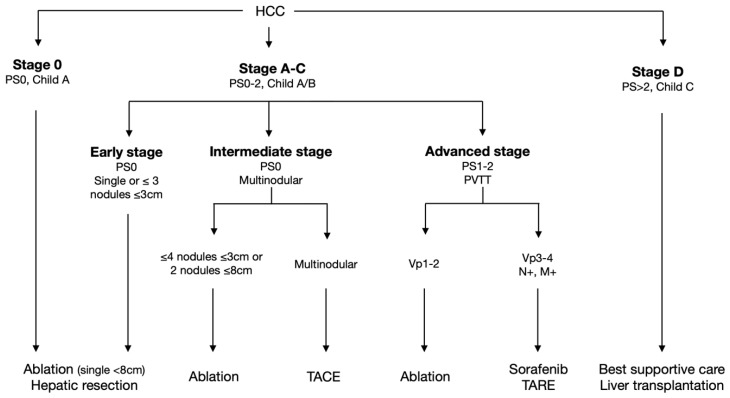
Protocol for the management of hepatocellular carcinoma in cirrhotic patients applied at Jean Verdier Hospital University (Bondy group) in first-line setting. Ablation techniques used at Jean Verdier Hospital are monopolar radiofrequency, multi bipolar radiofrequency, microwave, intra-arterial ethanol injection and irreversible electroporation [15,16]. Large ablation with multi bipolar radiofrequency is performed up to 8 cm [17]. After one of the initial treatments, consecutive therapies could be managed by curative options (salvage transplantation, resection, or ablation) or palliative options (TACE, TARE, and Sorafenib). Vp1-2: presence of a tumor thrombus distal up to the second-order branches of the portal vein. Vp3-4: presence of a tumor thrombus in the first-order branches of the portal vein or in the main trunk of the portal vein. At Jean Verdier Hospital, HCC with Vp1-2 portal invasion, if possible, is treated by ablation using either multi bipolar radiofrequency or irreversible electroporation or intraarterial ethanol injection [16,18,19]. HCC, Hepatocellular carcinoma; PS, performance status; HR, hepatic resection; TACE, trans-arterial chemoembolization; TARE, trans-arterial radioembolization; BSC, best supportive care; N+, nodal metastasis; M+, extrahepatic metastasis.

**Figure 2 cancers-14-02634-f002:**
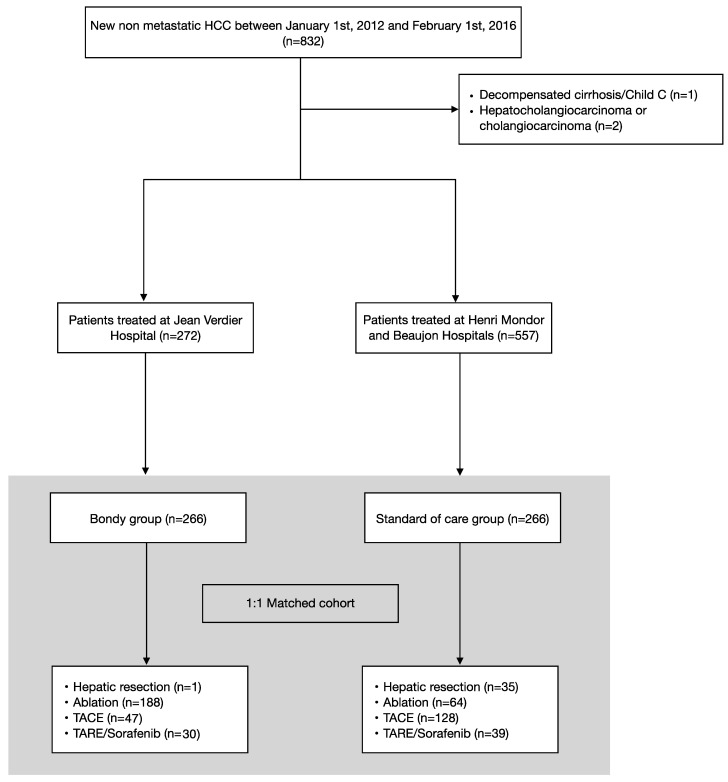
Consort diagram of study flow and the selection of patients for inclusion. The two groups were matched based on a 1:1 propensity score using an optimal matching program. HCC, Hepatocellular carcinoma; TACE, trans-arterial chemoembolization; TARE, trans-arterial radioembolization.

**Figure 3 cancers-14-02634-f003:**
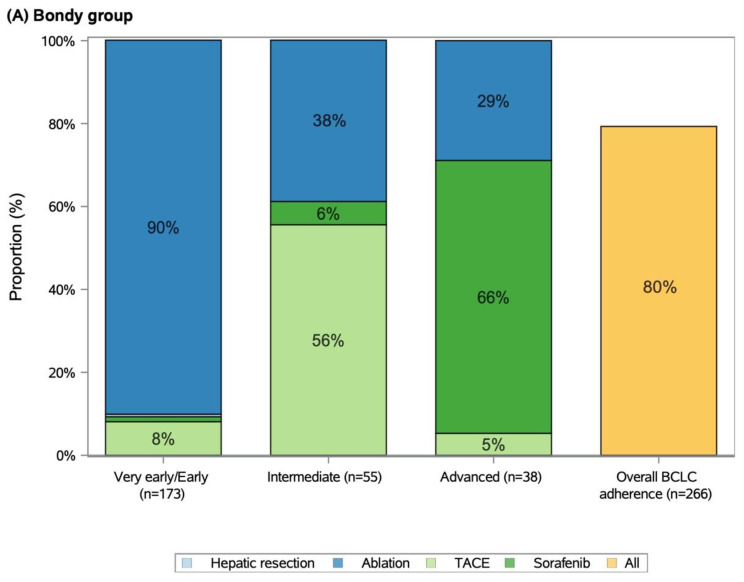
First-line treatment by BCLC stage in the matched (**A**) Bondy and (**B**) SOC group.

**Figure 4 cancers-14-02634-f004:**
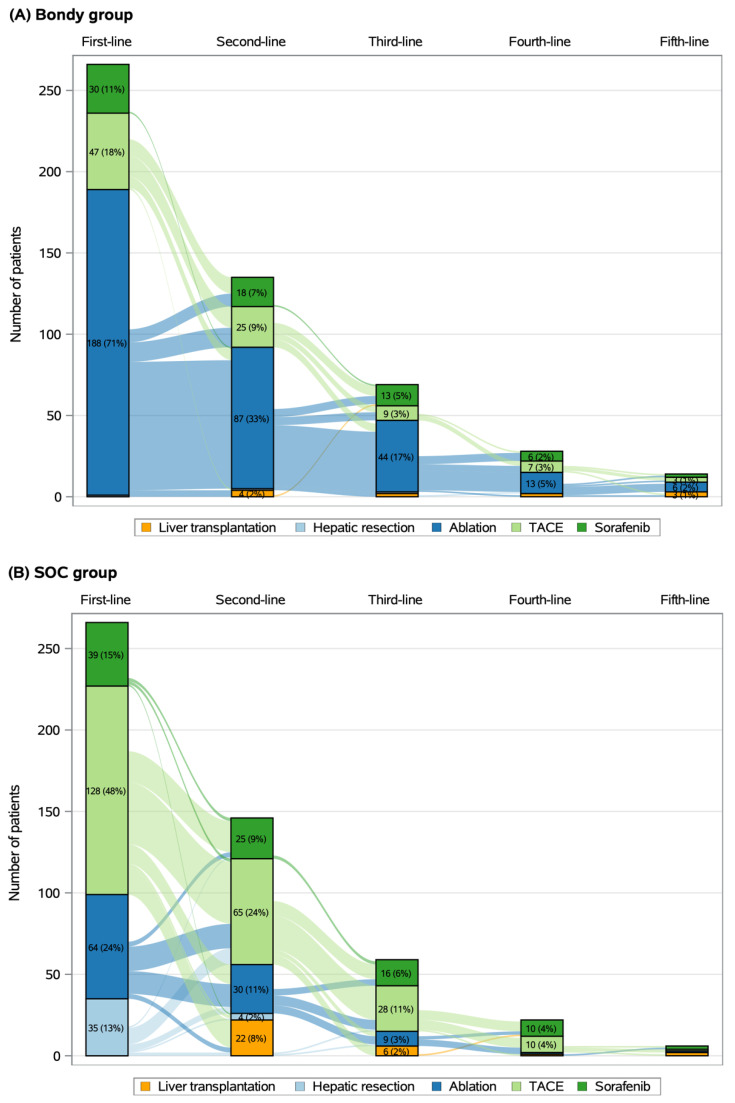
Therapeutic trajectories after first-line treatment in the matched (**A**) Bondy and (**B**) SOC group. The flows between bars are proportional to the number of patients at each step.

**Figure 5 cancers-14-02634-f005:**
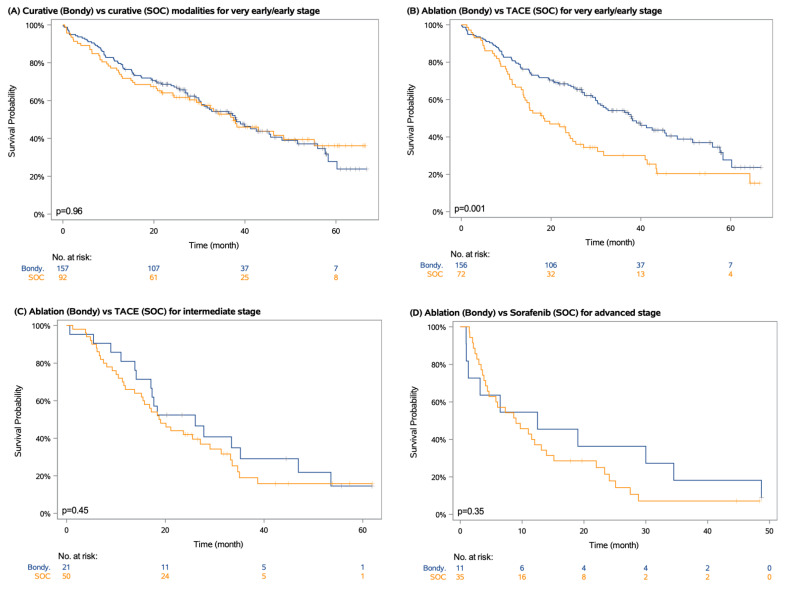
Kapan–Meier survival estimates by BCLC stage and first-line treatment. (**A**) very early/early HCC treated with by first-line curative options, (**B**) very early/early HCC treated with first-line ablation in the Bondy group versus first-line TACE in the standard of care group, (**C**) intermediate Child A HCC treated with first-line ablation in the Bondy group versus first-line TACE in the SOC group and (**D**) advanced HCC treated with first-line ablation in the Bondy group versus Sorafenib in the SOC group.

**Figure 6 cancers-14-02634-f006:**
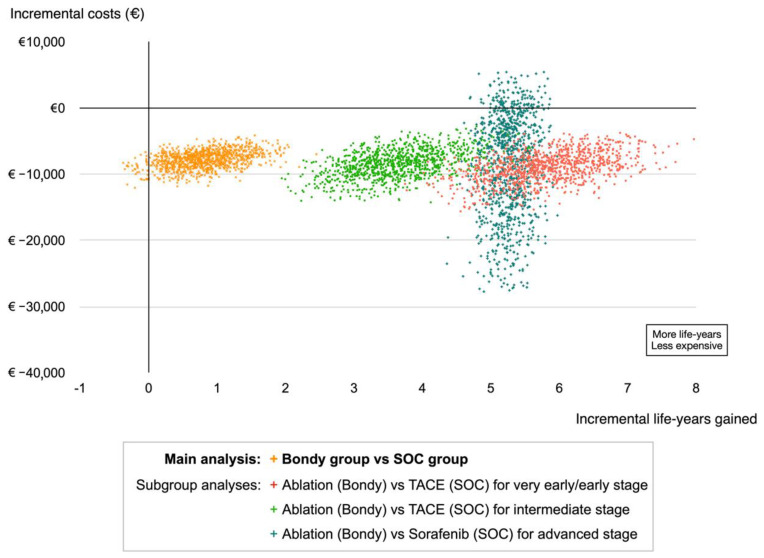
Scatter plot of the probabilistic sensitivity analysis from the matched cohort. The main analysis (Bondy group vs. SOC group) is depicted in orange. Subgroup analyses by BCLC stage are also presented. In the main analysis, 96.2% of the 1000 simulations were in the lower right quadrant (dominant strategy).

**Table 1 cancers-14-02634-t001:** Baseline characteristics of the unmatched and matched cohort and treatment groups.

Baseline Characteristics	Unmatched Cohort			Matched Cohort		
	Bondy Group	SOC Group	*p*-Value	Bondy Group	SOC Group	*p*-Value
	*n* = 272	*n* = 557		*n* = 266	*n* = 266	
Age, mean (+/−SD)	66 (+/−11)	63 (+/−10)	<0.001	66 (+/−11)	66 (+/−10)	0.37
Male, *n* (%)	219 (81%)	464 (83%)	0.32	214 (80%)	211 (79%)	0.75
Co-morbidity index, *n* (%)			0.33			0.09
≤1	262 (96%)	528 (95%)		256 (96%)	247 (93%)	
>1	10 (4%)	29 (5%)		10 (4%)	19 (7%)	
Cause of cirrhosis, *n* (%)			0.02			0.87
Alcohol	118 (43%)	190 (34%)		113 (42%)	117 (44%)	
HCV	89 (33%)	227 (41%)		89 (33%)	88 (33%)	
HBV	34 (13%)	83 (15%)		34 (13%)	32 (12%)	
NASH	20 (7%)	40 (7%)		20 (8%)	22 (8%)	
Others	11 (4%)	18 (3%)		12 (5%)	8 (3%)	
Child-Pugh score, *n* (%)			0.14			0.88
A (5–6)	249 (92%)	491 (88%)		243 (91%)	244 (92%)	
B (7–9)	23 (8%)	66 (12%)		23 (9%)	22 (8%)	
AFP level, *n* (%)			0.70			0.45
≤100 ng/mL	191 (70%)	398 (71%)		189 (71%)	178 (67%)	
100–1000 ng/mL	50 (18%)	97 (17%)		48 (18%)	50 (19%)	
>1000 ng/mL	31 (11%)	62 (11%)		29 (11%)	38 (14%)	
Portal hypertension, *n* (%)	158 (58%)	297 (53%)	0.20	154 (58%)	146 (55%)	0.48
Size of the largest tumor node, *n* (%)			0.42			0.29
≤30 mm	116 (43%)	260 (47%)		115 (43%)	119 (45%)	
30–60 mm	91 (33%)	181 (32%)		90 (34%)	93 (35%)	
>60 mm	65 (24%)	116 (21%)		61 (23%)	54 (20%)	
Number of tumors, *n* (%)			0.12			0.57
1	170 (63%)	314 (56%)		165 (62%)	166 (62%)	
2/3	63 (23%)	167 (30%)		63 (24%)	61 (23%)	
≥4	39 (14%)	76 (14%)		38 (14%)	39 (15%)	
Bilobar involvment, *n* (%)	65 (24%)	146 (26%)	0.47	64 (24%)	64 (24%)	1
Vascular invasion, *n* (%)	42 (15%)	63 (11%)	0.09	38 (14%)	43 (16%)	0.55
BCLC staging, *n* (%)			0.01			0.57
Very early	16 (6%)	60 (11%)		16 (6%)	22 (8%)	
Early	159 (58%)	290 (52%)		157 (59%)	143 (54%)	
Intermediate	55 (20%)	144 (26%)		55 (21%)	58 (22%)	
Advanced	42 (15%)	63 (11%)		38 (14%)	43 (16%)	
First-line treatment, *n* (%)			<0.001			<0.001
Hepatic resection	1 (1%)	69 (12%)		1 (<1%)	35 (13%)	
Ablation	190 (70%)	144 (26%)		188 (71%)	64 (24%)	
TACE	48 (18%)	283 (51%)		47 (18%)	128 (48%)	
TARE or Sorafenib	32 (12%)	60 (11%)		30 (11%)	39 (15%)	

Percentages may not total 100 due to rounding. SD, standard deviation; HCV, Hepatitis C virus; HBV, Hepatitis B virus; NASH, non-alcoholic steatohepatitis; AFP, alfa-fetoprotein; BCLC, Barcelona Clinic Liver Cancer; TACE, trans-arterial chemoembolization; TARE, trans-arterial radioembolization.

**Table 2 cancers-14-02634-t002:** Baseline results of cost-effectiveness analyses by first-line treatment. These results were weighted by the probabilities of each first-line treatment in the Bondy and standard of care groups and used to calculate the final ICER.

**Probability of First-Line Treatment**		**Bondy Group**		**SOC Group**	
Hepatic resection		0.4%		13.2%	
Ablation		70.7%		24.0%	
TACE		17.7%		48.1%	
Sorafenib		11.2%		14.7%	
**First-Line**	**Costs and Life-Years Per Patient**	**Bondy Group**	**SOC Group**	**Net Effects**	**ICER (€/LYG)**
Hepatic resection	Cost	€127 ($140)	€4417 ($4860)	€−4290 ($−4720)	
	Life-years	0.1	2.6	−2.5	
Ablation	Cost	€14,488 ($15,937)	€4904 ($5394)	€9584 ($10,543)	
	Life-years	11.1	3.8	7.3	
TACE	Cost	€2268 ($2495)	€13,478 ($14,826)	€−11,210 ($−12,331)	
	Life-years	0.7	4.3	−3.6	
Sorafenib	Cost	€1322 ($1454)	€2694 ($2963)	€−1372 ($−1509)	
	Life-years	0.2	0.5	−0.3	
Average total	Cost	€18,205 ($20,026)	€25,493 ($28,043)	€−7288 ($−8017)	Dominant
	Life-years	12	11.2	0.8	

Costs and life years are expressed as average for a patient and were discounted by 3%. The net effect could be a reduction in costs (negative values) or an increase in costs (positive values). TACE, trans-arterial chemoembolization; ICER, incremental cost-effectiveness ratio; LYG, life-years gained.

**Table 3 cancers-14-02634-t003:** Input parameters values, distribution, and boundaries for probabilistic sensitivity analyses.

Parameters	Distribution	Mode	Minimum	Maximum
Health state cost	Gamma			
Hepatic resection		€17,666 ($19,432)	€16,900 ($18,590)	€18,300 ($20,130)
Ablation		€4895 ($5385)	€4400 ($4840)	€5000 ($5500)
TACE		€5708 ($6279)	€5500 ($6050)	€5800 ($6380)
Sorafenib		€4500 ($4950)	€2935 ($3229)	€10,320 ($11,352)
Liver transplantation		€51,779 ($56,957)	€48,900 ($53,790)	€54,000 ($59,400)
Follow-up after hepatic resection or ablation		€146 ($161)	€100 ($110)	€500 ($550)
Follow-up after TACE		€379 ($417)	€100 ($110)	€800 ($880)
Follow-up after liver transplantation		€1514 ($1665)	€594 ($653)	€2000 ($2200)
Start age	Triangular	65	60	67
Transition probabilities	Beta	Original value	10% lower	10% higher
Discount rate	Triangular	3%	0%	6%

## Data Availability

The data presented in this study are available on request from the corresponding author.

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
