# Peer review of "Impact of Extended Use of Ablation Techniques in Cirrhotic Patients with Hepatocellular Carcinoma: A Cost-Effectiveness Analysis"

_cancers, 2022, doi:10.3390/cancers14112634_

Round 1
Reviewer 1 Report
Dear authors, I have read again this paper that you have now improved according with reviewers' suggestions. These suggestions were addressed only with the aim to make your messages more clear, that is, to make the reader of Cancers capable to enjoy your study. What is obvious for you as authors (see point 13 of your reply) might not be for the general readership.
This manuscript is a resubmission of an earlier submission. The following is a list of the peer review reports and author responses from that submission.
Round 1
Reviewer 1 Report
I have read with interest this manuscript, which concerns the role of what the authors called extended ablation technique in patients with HCC and cirrhosis.
The aim of this paper was to provide the cost-effectiveness analysis by comparing the so called extended ablation techniques versus the non-extended ablation techniques in HCC patients.
While I believe that the paper is overall of interest, I think that this manuscript should be carefully revised and rewritten because of many misconceptions.
- The authors take for granted the meaning of extended and non-extended ablation techniques, which by the way are not. Such definition together with much more technical details should be provided.
- Please remove the number on the paragraphs in the abstract.
- Please define all the provided acronyms starting from those in the abstract.
- Figure 1 shows a revised version of the BCLC classification. No references are available for that. If this is what authors do in their centers, the authors should explain much more this classification. Indeed, there is very small room for resection and for transplantation, which should not be confined in stage 0 and in stage D respectively. Or, if the authors have date to support these indications, they should write down at least some specific references.
- The performance of ablations in patients with multinodular HCC is not new, but is always debated. The categories of tumor number proposed are new and they should be weighted on previously published papers.
- Similarly, the performance of ablation in patients with Vp1-2, meaning in patients with PVTT in a given segmental or sectorial portal branch, requires some justifications, some data that again should be given to the reader. In general, the ablation of a HCC in S8 with PVTT in P8d is not a good indication. This is valid also when the same patient cannot be operated or transplanted.
- The text from row 97 to 102 belongs to figure legend 1.
- Please provide more definitions. How was cirrhosis defined? Histologically?
- How was portal hypertension defined?
- Since 3 patients were excluded because of mixed histology, the reader might think that all the included patients had liver and tumor biopsy performed before the ablation. Is this the case? In general, this is not and this is one of the main limitations of such a study in comparison with studies in which any included patient had a formal histology diagnosis (i.e. patients treated with surgery).
- It is unclear if those included patients had ablation as first diagnosis and first therapy of their HCC.
- Figures 2 - 4 are almost unreadable.
- Finally, it is unclear if the research question here is about the technical feasibility of extended techniques proposed by the Bondy group or the associated oncological outcomes or maybe both or the costs.
- I would suggest to clarify much better the study inclusion/exclusion criteria together with adding some missing definitions and clarify the research question.
Reviewer 2 Report
The authors prepared a valuable manuscript on a question of high clinical and socioeconomic impact. Treatment of HCC, though more or less standardized by EASL/BCLC, underlies strong variability among centres. However, extended use of local ablative curative techniques, whenever feasible, show superior benefit or better tolerability as compared to surgery, locoregional treatments or palliative care. The presented data fortifies the argumentation pro local ablative approach by analysing cost effectiveness of an “extended local ablative approach”. The submitted work indicates that it is more effective and less expensive to offer cirrhotic patients with HCC an extended use of ablation treatments, particularly at earlier stages of the disease. Further on, from a clinical perspective very interesting, a substantial proportion of patients treated had a shift from palliative to curative treatments by the extended approach, underpinning the need for redefinition of treatment recommendations.
The article is well written, methods are clearly described, statistics and models are appropriate, discussion position the findings appropriately in the scientific background, limitations are well addressed.
Regarding extended local ablative approach I would like suggest to include the technique of catheter based radiotherapy (brachytherapy) in the introduction/discussions. As with the techniques described by the authors (by extended use of thermoablative techniques), with brachytherapy the borders of regular thermal ablations can be extended. A prospective randomized trial comparing brachytherapy to TACE showed a significant longer time to untreatable progression in the brachytherapy arm. OS data did not differ between the arms, most probably due to an allowed cross over between the arms (TACE to brachytherapy in TACE refractoriness). Please add information and refer to PMID: 30488303